# Job Satisfaction Among Healthcare Professionals in Community-Based Care for Older People: Evidence from Greece

**DOI:** 10.3390/healthcare13182299

**Published:** 2025-09-13

**Authors:** Eirini Michaela Foukaki, Argyroula Kalaitzaki, George Markakis, Athanasios Alegakis, Sofia Koukouli

**Affiliations:** 1Heraklion Mental Health Center, Venizeleio Hospital, 71305 Heraklion, Greece; 2Department of Social Work, Hellenic Mediterranean University (HMU), 71410 Heraklion, Greece; akalaitzaki@hmu.gr (A.K.); gmark@hmu.gr (G.M.); koukouli@hmu.gr (S.K.); 3Quality of Life Lab, Faculty of Health Sciences, Hellenic Mediterranean University, 71410 Heraklion, Greece; 4University Research Centre Institute of AgriFood and Life Sciences, Hellenic Mediterranean University, 71410 Heraklion, Greece; 5Lahers, Laboratory of Health and Road Safety, Hellenic Mediterranean University, 71410 Heraklion, Greece; 6Department of Toxicology, Faculty of Medicine, University of Crete, 71003 Heraklion, Greece; alegkaka@uoc.gr

**Keywords:** job satisfaction, aged, open care services, health personnel, Greece

## Abstract

Objectives: This study aimed to investigate the level of job satisfaction and the associated factors among health and social care employees of the public sector providing services in open care community settings and home-based care for the older people in Greece. Method: The self-administered questionnaire, Job Satisfaction Survey (JSS), was distributed to personnel across all four prefectures in the Region of Crete (Greece). In total, 228 valid responses were received. Results: Overall job satisfaction was moderate (mean = 134, SD = 24). Employees reported greater satisfaction with the supervision (mean = 19.3), nature of work (mean = 19.2), and coworkers (mean = 19.0). Lower satisfaction was observed in relation to promotion (mean = 9.7). Women scored significantly higher satisfaction than men in pay (mean = 10.9) and fringe benefits (mean = 12.3), while age was associated with differences in pay and promotion satisfaction. Variations were also found according to service unit and contract type, with permanent staff reporting higher satisfaction (mean = 11.3) in the Promotion scale than temporary staff (mean = 9.2) (*p* < 0.001). Conclusions: Findings emphasize the need to institute targeted short- and long-term measures to improve job satisfaction in community-based care for older people. Short-term actions should include the introduction of fair and competitive pay structures, improvements in fringe benefits, and the implementation of supportive leadership practices. Meanwhile, long-term strategies should focus on transparent promotion systems, structured professional development and continuing education opportunities, and the provision of stable contracts to retain skilled personnel.

## 1. Introduction

The demand for senior care has increased worldwide due to the rise in the aging population [1]. In 2023, the proportion of adults aged 65 and older in the EU was 21.3%, reflecting an increase of 0.2% compared to 2022 and of 3.0% from a decade prior [2]. This demographic shift is putting pressure on healthcare and social care systems to provide a continuous spectrum of services that meet physical, psychological, and social needs. The care network for older people typically combines informal care—unpaid support from family members, relatives, neighbors, and friends—with formal care delivered by professionals in residential, community-based, or home-based settings [3,4]. However, as the traditional informal care is unable to satisfy the increasing need, governments are prioritizing community-based and home-based care as sustainable alternatives to institutionalization [5,6]. Furthermore, available evidence indicates that community care, including home-based care, has a positive impact on the social, psychological, and physical health of older adults, enhances their quality of life [5,6,7], and delays entry into residential care [8].

In Greece, due to traditionally strong family values, combined with limited formal services, informal home-based care was the most predominant type of care for older people [9]. Although there are no official or robust figures on the absolute number of informal carers in the country, an estimated 14–30% of adults aged 16 years and older provide informal care based on survey data from 2016 and 2022 [10,11]. However, the increasing participation of women in the labor market and changes in family structure have contributed progressively to the significant reduction in informal care networks in recent years. To address this gap, three main public sector programs—Home-Based Care (BSS), Open Care Centers for the Older People (KAPI), and Day Care Centers for the Older People (KIFI)—play a vital role in meeting the needs of older adults in the community. The effectiveness and sustainability of these services depend largely on the job satisfaction of their workforce, as professional well-being is directly linked to service quality [12,13,14,15,16] and to the psychological and physical health of employees themselves [17,18].

Job satisfaction—one of the most significant and frequently studied variables in the field of organizational psychology [19,20]—is generally defined as a positive emotional response that arises from an evaluation of one’s job or work experiences [21,22,23].

Several researchers have classified job satisfaction predictors into three categories: individual, relational, and organizational [14,24]. Individual variables include self-efficacy, self-esteem, or demographic features of employees. Relationships with employers, supervisors, and coworkers are examples of relational elements. Salary, opportunities, promotions, and various benefits belong to organizational features [14,24].

Different instruments have been developed to measure job satisfaction. Job Satisfaction Survey (JSS) is one of the most frequently used to this day [25]. Μany studies tested its psychometric properties [25,26,27,28,29]. Spector [30] summarized the variables that were found to relate to job satisfaction. These included turnover, intentions of quitting the job, perceived job characteristics, the leadership style, absenteeism, personal characteristics, pay, age, and organizational level [30].

The conceptual framework for this investigation was the Herzberg two-factor theory, also known as Motivation–Hygiene Theory. According to this theory, employee motivation is influenced by two sets of factors. One was categorized as “motivational,” and the other as “hygiene” [31]. Motivational factors intrinsic to the work itself include achievement, recognition, responsibility, and advancement. These factors motivate employees to work harder and perform to higher standards. “Hygiene” factors, or ‘dissatisfiers’, concern the job environment, including company policies, practices, remuneration, fringe benefits, working conditions, administration, supervision quality, interpersonal relations, salary, status, and security [31,32].

Studies on job satisfaction in healthcare services often refer to Herzberg’s theory [33,34,35,36]. Recent research in Greek healthcare settings [37] confirms the applicability of Herzberg’s model in studying job satisfaction among healthcare professionals, highlighting its importance in community care settings. The JSS [30] implements this dual theory by assessing nine aspects that include both intrinsic motivators (e.g., nature of work, promotion, and recognition) and extrinsic hygiene factors (e.g., compensation, supervision, and operational procedures).

Although job satisfaction has been widely studied in hospital-based settings [1,13,14,18,38], considerably less attention has been paid to professionals working in community-based care for older people, and very few studies have compared satisfaction levels across different service types. The gap is especially significant in the Greek context due to the variety of programs (BSS, KAPI, KIFI) and the growing dependence on these services to address demographic and policy concerns.

This study seeks to address this gap by investigating intrinsic and extrinsic factors of job satisfaction, as well as comparing job satisfaction levels among employees in the three primary community-based aged care services in Crete.

## 2. Materials and Methods

### 2.1. Aim

The study’s objectives were (a) to investigate the degree of overall job satisfaction among health and social care professionals employed in open care services for older people; (b) to examine intrinsic motivators and extrinsic aspects of their job satisfaction; (c) to compare job satisfaction levels across three types of community-based services; and (d) to examine the effects of personal and work characteristics on job satisfaction,

### 2.2. The Design of the Study

The present research was a cross-sectional quantitative study conducted between June 2022 and March 2023.

### 2.3. Participants and Setting

The study was conducted at three public sector services that provide open-based care for the older people in the Regional Unit of Crete in Greece. This research was designed as a case study focusing on the Region of Crete, Greece. Given that Crete is an island, its geographical and demographic characteristics may influence the organization and delivery of community-based elder care services, potentially differing from those in mainland regions. Crete is the biggest island in Greece and one of the 13 regions of the country. According to the 2021 Census, the Regional Unit had a total population of 624,408 inhabitants and 154.983 people 60 years and over (24.8%) [39]. These three services were KAPI (Open Care Centers for Older People), KIFI (Day Care Centers for Older People), and the Help at Home Program (BSS). In the region under investigation, there were 42 KAPIs, 16 KIFIs, and 67 BSS programs during the study period.

KAPI (the Greek acronym for Open Care Centers for Older People) serves as a primary social policy service for autonomous older people in Greece. They work under municipal institutions as part of the national welfare system. The Center provides prevention services and addresses needs in the context of open and non-institutional care, aiming to assist senior citizens during the sensitive phase of old age and the accompanying physical, psychological, and social changes. The purpose of this service is: (a) to implement programs that contribute to older people staying in the familiar environment of their home and to be functional, active, and useful to society as a whole; (b) to be a source of support for lonely and disabled people as well as a source of information for families caring for older people or disabled (older) members; and (c) to educate society about aging, identify the needs of older people and assess the community’s sources of empowerment. Therefore, the main goal is to ensure conditions of care, protection, information, development, entertainment, and, in general, to maintain older citizens as equal and active members within society. Attendance is voluntary, without invitation, and older adults may participate in activities for as long as they wish during the center’s operating hours, typically in the morning and early afternoon.

KIFI (the Greek acronym for Day Care Centers for Older People) provides daily accommodation services for older individuals who are unable to take care of themselves due to mobility difficulties, health problems, dementia, etc. Moreover, their family caregivers cannot fulfill their care responsibilities since they are either working or dealing with serious social, financial, or health issues themselves. The services operate in specially designed spaces on a daily basis and can accommodate older people for 4–8 h a day, depending on their needs and service availability, with attendance arranged in agreement with family caregivers, providing care services (daily hygiene and nursing), entertainment, and psychosocial support.

BSS (the Greek acronym for Help at Home Program) aims to provide home services to older adults and people with disabilities who mainly live alone and whose income is severely low. Operating as part of municipal social care policy, its primary goals are to enable beneficiaries to remain in their familiar, natural, family and social environment and to avoid social exclusion. Services can involve support for everyday duties, visits to the home by care experts, and connections to community resources.

The participants were the employees of the local government’s open care services for the older people (KAPI, KIFI, and BSS), located in all four prefectures within the Region of Crete. Participants’ selection was based on the two following predefined inclusion criteria: (1) being an employee with over one year of work experience; and (2) being employed at the three services of the public sector that take care of older people in an open base at the community in the Region of Crete in Greece. The exclusion criteria were two: (1) employees on long-term leave during the data collection period; and (2) employees unwilling to participate.

The participants of the present study include members of the healthcare teams of various specializations (social workers, nurses, social caregivers, support personnel, etc.), educational backgrounds, and employment status (permanent or temporary staff). All specialties were included in the study to ensure a representative overview of the workforce employed in these services. Different specialties contribute at different levels to the daily operation and care of older people; therefore, their participation in the research offers a holistic perspective on the demands and challenges encountered in these settings.

To guarantee sufficient statistical power and representation across all services and prefectures, a percentage between 50% and 60% of all employees across the three services was determined. Following the mapping of services, it was found that there were 447 employees at the time of the study. A sample size of 250 participants was established based on this desired percentage. Of these, 228 workers consented to participate and sent back their filled-out surveys, resulting in a 91% response rate.

### 2.4. Measures

A self-administered questionnaire was distributed to the selected employees working in the services mentioned above. The questionnaire included two parts:Part A: The socio-demographic profile of the participant

The socio-demographic items measured in our study were as follows: sex, age, educational level, marital status, number of children, region of work, place of residence (rural or urban area) service in which participants are employed, employment status, participants’ specialization, years of work experience, whether or not the participant held a position of responsibility, professional title, and monthly income.

b.Part B: The measurement of job satisfaction

The survey tool selected in this research was the “Job Satisfaction Survey” (JSS) created by Spector [30] and translated and revised in Greek by Tsouni A. and Sarafi P. [25]. Additionally, the Greek translation of the Job Satisfaction Survey (JSS) has been validated for both validity and reliability. Validity: Confirmatory Factor Analysis supported the original nine-factor structure, with good model-fit indices (RMSEA = 0.055, CFI = 0.951, GFI = 0.946) and strong factor loadings (0.61–0.90).Reliability: Cronbach’s alpha coefficients for the subscales ranged from 0.62 to 0.87 (with the exception of Operating Procedures, α = 0.48), while the overall scale achieved α = 0.87. Split-half reliability was also strong (Guttman = 0.876, Spearman–Brown = 0.877). The fact that the tool has been designed for use primarily in psychosocial care and social service organizations in the public sector made it perfectly suitable for the present study.

The questionnaire has a rating scale from one to six (1 = strongly disagree, 2 = moderately disagree, 3 = slightly disagree, 4 = moderately agree, 5 = slightly agree, and 6 = strongly agree). It is easy to process, includes 36 questions, and examines nine dimensions (with four items each) related to job satisfaction as mentioned below: (1) Pay: Unfair distribution of pay can negatively affect employees’ emotions and behavior, e.g., “I feel I am being paid a fair amount for the work I do”. (2) Promotion involves progression to higher positions with more challenges and responsibilities, e.g., “There is really too little chance for promotion on my job”. (3) Supervision concerns the employees’ perception about the support received from supervisors, e.g., “My supervisor is quite competent in doing his/her job”. (4) Fringe benefits include financial and non-financial compensations, such as bonuses and retirement plans., e.g., “I am not satisfied with the benefits I receive”. (5) Contingent rewards are valuable tools for motivating employees, e.g., “When I do a good job, I receive the recognition for it that I should receive”. (6) Operating procedures are essential for achieving goals and maintaining a positive work environment, e.g., “Many of our rules and procedures make doing a good job difficult”. (7) Co-workers with similar values and attitudes can improve satisfaction and reduce stress, e.g., “I like the people I work with”. (8) The nature of work, including challenges, feedback, autonomy, and skill variety, can increase motivation and internal happiness, ultimately leading to satisfaction, e.g., “I sometimes feel my job is meaningless”. (9) Communication: there is a positive association between employees’ and managers’ communication and job satisfaction, e.g., “Communications seem good within this organization” [40].

Scores on each of the nine-facet subscales can range from 4 to 24, and scores for total job satisfaction, which are based on the sum of all 36 items, can range from 36 to 216. JSS has 19 negatively worded items, which must be reversed [30]. The score for overall work satisfaction ranges from 36 to 216 (where 36–108 indicates dissatisfaction and 144–216 satisfaction, while the values 108–144 range in the area of neutrality) [17]. Numerous studies have shown that the JSS has high internal consistency and validity [40].

### 2.5. Data Collection

Data collection started immediately after the revocation of Greece’s COVID-19 restrictive measures, as the three services were obliged to modify their operational methods during the pandemic. The study was conducted following official approval from the participating municipalities. The researcher contacted the managers of the structures to explain the goal of the study and to provide them with all the necessary information. Service managers informed eligible employees about the study’s purpose and procedures, and participation was voluntary. The participants were asked whether they preferred to fill out the questions online using Google Forms or on paper. All of them chose to fill out the questionnaires by hand. Thus, the questionnaires were distributed in two ways: either by email or by regular mail in sealed envelopes. Mailed questionnaires were also sent in postage-paid envelopes so that the completed questionnaires could be sent to the researcher without burdening the participants. Two phone calls were made: one to confirm that the questionnaire was received, and another to verify that the researcher had received it. Three weeks was the average waiting time, and in some instances, more than three phone calls were required to obtain the completed questionnaires. Questionnaires were collected anonymously.

### 2.6. Ethical Considerations

The research protocol was initially approved by the Department of Social Work of the Hellenic Mediterranean University (Protocol Number 2989/5-5-2020) and by the Ethics Committee of the University (Protocol Number 11715/16-7-2024). In order to conduct the research, 28 letters of approval were sent to the implementing and management bodies of the programs related to the field of study (municipalities, public beneficiaries, municipal enterprises, and municipal social policy practice departments). The procedure was then differentiated for each municipality. Only a few municipalities responded immediately, either in writing or via telephone communication, while in most other cases, repeated telephone communication was required. In some cases, further information was requested before their consent, such as the detailed research protocol. Finally, there were two municipalities that refused to participate in the investigation.

### 2.7. Statistical Analysis

The IBM SPSS Statistics 24.0 program was used for data classification and statistical analysis. In addition, the continuous variables, such as the calculated scores of the scales, were expressed in the form of a mean and standard deviation. Discrete variables were expressed as frequency and % frequency.

The distribution of calculated scores was checked for the regularity of their distribution with the Kolmogorov–Smirnov test. Depending on the data normality or not of the distribution, differences were examined using parametric or non-parametric tests, respectively. In the case of mean differences between two groups, the independent sample *t*-test or the corresponding non-parametric Mann–Whitney test was used. In the case of comparisons between two or more groups, a one-way ANOVA or the corresponding non-parametric Kruskal–Wallis test was used.

The correlation between two continuous variables was examined with Pearson’s or Spearman’s r coefficients, while the correlation of discrete variables was examined with Pearson’s χ^2^ test. Scatterplots, barcharts were used for data graphical representation.

A *p*-value < 0.05 was set as the level of significance.

## 3. Results

The sample consisted primarily of females (89%), with a mean age of 47.2 years, and over half belonged to the 45–55 age group. The majority of participants were married and resided in urban locales. The predominant educational qualification was higher education, with most individuals indicating monthly wages ranging from €801 to €1500. Table 1 presents comprehensive demographic characteristics.

The data in Table 2 show that the majority of participants work on temporary contracts, which indicates limited job security. At the same time, the majority hold staff positions rather than managerial roles, although most have many years of professional experience; almost six out of ten have been working for over 15 years. This information reveals a contradiction between long-term experience and limited professional recognition. Similarly, at the level of community service programs, almost half are employed in the BSS, while KIFI and KAPI account for smaller but comparable percentages, highlighting the central role of the BSS in the structure of services.

The distribution of specializations, as shown in Figure 1, indicates that most workers are employed in three main roles, nurses, social workers, and home health aides, who collectively account for over 70% of the total. The remaining specialties appear at much lower percentages, each below 10%. This image illustrates a staff structure that emphasizes core caregiving professional roles, while specialties such as social workers, support staff, and therapists work complementarily.

Participants’ overall satisfaction, as measured by the JSS scale, is around the average value (mean = 134), indicating moderate overall job satisfaction. Regarding the individual dimensions, the highest average scores are observed in the areas of supervision, nature of work, and colleagues, indicating that employees are relatively satisfied with the guidance they receive, their own work, and their collaboration with colleagues. Conversely, the lowest average scores are recorded in the dimensions of promotion and fringe benefits, indicating dissatisfaction with career advancement prospects and additional perks. The distribution of the results does not follow a normal distribution, which should be taken into account in further analyses (Table 3).

The effect of sex on overall job satisfaction and the JSS subscales is presented in Table 4. In most subscales, women reported numerically higher satisfaction than men, except for the ‘Communication’ subscale. However, statistically significant differences were found only in pay and fringe benefits subscales.

Age appeared to influence specific aspects of job satisfaction, as measured by the JSS (Table 5). The highest satisfaction in the pay subscale was observed in the 46–55 age group, compared to participants aged ≤45 and ≥56, with a statistically significant difference. Also, participants 46–55 reported higher satisfaction in the promotion subscale compared to those under 45 and those aged ≥56, also showing a significant difference. These results suggest that middle-aged employees perceive higher pay satisfaction, while younger employees are relatively less satisfied with promotion opportunities.

Figure 2 depicts the correlation between years of professional experience and the JSS subscales. A statistically significant positive connection was identified between years of experience and the pay subscale. A trend indicating a positive link was observed for the promotion subscale, although it did not achieve statistical significance. The findings indicate that employees with greater experience generally report marginally higher satisfaction with pay, but promotion satisfaction exhibits a comparable, though non-significant, trend.

Accordingly, Figure 3 illustrates the relationship between net monthly income and the JSS subscales. A notable positive link was identified between net monthly income and the pay subscale, suggesting that increased income correlates with enhanced satisfaction with pay. A notable negative connection was identified between net monthly income and the operational procedures subscale, indicating that employees with more income tend to report marginally poorer satisfaction with the way the organization operates.

The JSS subscale scores were compared to the type of work unit (KIFI, KAPI, and BSS) (Table 6). In the fringe benefits subscale, BSS employees reported the highest satisfaction, followed by KAPI and KIFI, with significant differences. Furthermore, the operating procedures subscale demonstrated substantial disparities among work divisions, with KIFI employees expressing the highest levels of satisfaction in comparison to KAPI and BSS. These results emphasize that the nature of the work unit can significantly impact specific aspects of job satisfaction, particularly those related to operational procedures and fringe benefits. This implies that organizational interventions could be customized to meet the unique needs of each unit to enhance employee satisfaction.

Additionally, the JSS promotion satisfaction scale differed statistically significantly between employees with a permanent relationship (11.3 ± 4.2) and employees with some type of contract (9.2 ± 4.0) (*p* < 0.001). All other JSS scales did not differ significantly (*p* > 0.05).

Education, place of residence (rural or urban), and marital status did not show any statistically significant effect on any of the JSS job satisfaction scales.

In a regression model the total JSS score was tested for association from demographic (sex, age, educational levels, marital status, economical status and occupational factors (working unit, work experience, specialty (healthcare or not) and work position (head/director or not)) None of the variables showed a significant effect on the JSS score (ANOVA F (11, 220) = 0.830, *p* = 0.610, R^2^ = 0.042).

Each of the JSS subscales was tested for possible associations with demographic and occupational data. Multiple linear regression models for each subscale were tested using forward selection. In Table 7 and Table 8, the beta coefficients from the explanatory variable on each JSS scale are shown. Three of the JSS scales do not show any converge on the explanatory variables (contingent Rewards, Coworkers, Communication). R^2^ values were less than 0.100 in all models. Age groups, sex, monthly income, working status, working unit, and working position were the factors shown to be included in JSS subscale models.

## 4. Discussion

This study used Spector’s [30] developed questionnaire to assess job satisfaction in the public sector’s open services for older people. The main objective of the study was to investigate overall job satisfaction, identify key factors affecting it, and explore variations in satisfaction across employees working in the three different services (KAPI, KIFI, BSS). According to the study’s objectives, we reached the following observations.

Overall, job satisfaction across all three services was moderate (within the neutral range), a finding that is in line with those of other similar studies in Greece [17,41,42,43,44]. Although none of the groups reached levels indicating high satisfaction, some variations between them were observed. Specifically, besides the co-workers subscale, the two subscales with the highest mean scores were supervision and the nature of work. These findings are consistent with other studies both in Greece and abroad [17,25,41,42,43,44] b underlining the services’ collaborative and human-centered nature. The finding can be elucidated by the fact that the three services examined in the research were developed locally and are overseen by the local authorities. This fosters proximity and regular communication between employees and supervisors. According to studies on employee satisfaction in community services for older people, the staff members seem satisfied with their jobs since they express pride in their work and regard their choice as commendable [45,46]. Conversely, the subscales measuring fringe benefits and promotion were at the lowest levels, a finding consistent with other research in Greece [44]. This might be explained by the fact that in Greece’s public sector, there are very few opportunities for employees to receive promotion or fringe benefits.

The majority of the sample consisted of women (89.0%), a finding consistent with the fact that the occupational fields represented in all three structures under study remain female-dominated [47,48,49,50]. Moreover, in the case of BSS, one of the main reasons for its creation was the employment or integration of unemployed women into the labor market. Additionally, in the present study, women appeared more satisfied overall and across all subscales, with statistically significant differences in pay and fringe Benefits. It is well documented by a number of research studies on job satisfaction that women are happier than men despite the difficulties they are facing [40,51], a trend attributed to an intrinsic appreciation of the work and the achievement of a favorable work–life balance [51]. In the current context, these factors may be reinforced by the caregiving nature of elder care—traditionally associated with women—and by the fixed morning schedule of the services, which operate five days a week and remain closed on weekends and public holidays [52].

Subsequently, the following finding refers to the participants’ work experience. The long-term employment of the majority of participants (78.8% for over 11 years and 59.3% for over 16 years) indicates significant job stability. According to Herzberg’s Two-Factor Theory [31,32,37], job security is included among hygiene factors, which do not directly increase job satisfaction but mitigate dissatisfaction. Additionally, according to the findings of this research, a significant percentage of participants (47.7%) have only completed primary or secondary education, a fact that, according to the literature, can limit their prospects for professional development, thus strengthening their commitment to their current roles [53]. Moreover, in this study, extensive experience appears to strengthen motivational factors, such as positive relationships, as reflected in higher scores on the coworkers’ subscale, while increased satisfaction on the pay subscale—also a hygiene factor—is enhanced in the public sector by a hierarchical pay structure that rewards seniority.

Workers aged 46–55 reported higher satisfaction compared to younger and older age groups. Many other studies have also shown a relationship between age and job satisfaction [44,54]. Specifically, higher satisfaction is observed with pay and promotion—hygiene factors that contribute to maintaining a stable work environment [54]. This may be explained by the fact that senior employees, possessing greater experience, tend to feel more comfortable and confident in their current work conditions. Furthermore, while the absence of retirement-related concerns in this age group further enhances their satisfaction, conversely, older workers in Greece often face stagnation in their professional and economic development, while younger workers encounter significant obstacles in their advancement [55].

Regarding the relationship between job satisfaction and the type of work unit (KIFI, KAPI, and BSS), some intriguing results have emerged that can be interpreted within Herzberg’s Theory. Overall, employees across all three services exhibited neutral satisfaction, a finding consistent with prior studies on job satisfaction in the public sector in Greece [17,40,56]. However, the employees of KIFI showed the highest satisfaction, followed by those of BSS, while those of KAPI displayed the lowest. From a Herzberg perspective, such differences may reflect variations in the presence of motivators (e.g., meaningful work, recognition) and hygiene factors (e.g., working conditions, organizational support). Actually, this result can be interpreted by the fact that the KAPI are the first open care services established in the country for the protection of older individuals, and therefore the majority of the employees who participated in the survey as KAPI employees are workers who have been employed for over 20 years and are at the brink of retirement and experience reduced opportunities in career progression (an important hygiene factor). Another hypothesis that may explain these findings relates to the fact that these structures are permanent state institutions. Greece, from 2010 until recently, had severe restrictions on hiring personnel due to memorandum commitments, resulting in many significant shortages for these structures. Furthermore, the inability to fill these voids due to resignations and retirements affected the way these services operated.

When particular subscales were examined, the JSS fringe benefits scale revealed significant statistical differences, with the ΒSS employees demonstrating higher average scores compared to employees of the other two services. This finding coincided with a pivotal period for the employees of the Help at Home programs, who, after 20 years of working with renewable contracts in employment insecurity, managed to become permanent staff. This is a notable finding that confirms Herzberg’s Theory, mentioning that improving hygiene factors (job security, benefits) can prevent dissatisfaction.

The JSS operational procedures scale revealed a statistically significant difference among the three structures, indicating higher satisfaction among KIFI employees. This might be because, compared to the other two structures under study, KIFI receives daily a concrete number of beneficiaries and has a well-defined operational framework, which provides professionals with a sense of security. Conversely, the BSS program was designed to assist senior citizens in their homes, which can increase the unpredictability of the work environment. Additionally, the KAPIs beneficiaries may vary daily.

Finally, the research also indicated that employees with permanent job status had higher satisfaction regarding the promotion subscale compared to employees with some type of contract. As a result, it highlights the role of professional development as a key motivator in Herzberg’s model. This observation aligns with the prevailing circumstances in Greece, where employees bound by various forms of contracts experience significantly restricted professional rights, notably in terms of advancement opportunities.

## 5. Limitations

However, some limitations should be noted and taken into account. One limitation is the heterogeneity of the sample, which is related to their specialties and education level. In this particular study, the sample population employed in the services of interest comes from all four educational levels (Higher Education, Technological Education, Secondary Education, and Unskilled Labor), and from a wide range of specialties, which may influence their expectations and perception of job satisfaction. This diversity may limit the ability to generalize the findings to the broader sector.

In addition, the study was conducted exclusively in the Region of Crete, which is an island. The study showcases that distinct geographical, demographic, and service accessibility features might not accurately mirror the dynamics of elder care in mainland Greece. Therefore, caution is required when generalizing to other regions.

Another significant limitation concerns the methodological design of the research, which was based on the use of the structured JSS questionnaire. The use of a questionnaire is the least time-consuming process, allowing for data collection from a large sample size. On the other hand, it may limit the depth of responses and introduce response bias, such as a tendency toward socially desirable answers. The absence of qualitative methods restricted the ability to explore in more detail the underlying reasons behind the quantitative findings. Finally, a significant limitation during the writing of the article concerns the ability to compare the present results with those of other studies, given that two out of the three services (KIFI and KAPI) of interest do not exist in other countries, and difficulties are also encountered in comparing the results obtained with other research tools.

Despite the enumerated limitations, numerous advantages are linked to the current research. Initially, collecting data from a substantial, varied sample enhances the validity of the findings and provides an accurate representation of the present condition of these particular services. It improves comprehension of the expectations and perspectives of social services personnel, who have received less attention in the Greek setting. At the end, it is significant to note that in Greece, there is no other research investigating job satisfaction within these three services.

## 6. Conclusions

This study aimed to examine the characteristics influencing job satisfaction among personnel of healthcare teams employed in public open care services for older people in the Region of Crete, Greece, utilizing the Job Satisfaction Survey (JSS). The findings indicate that employees expressed satisfaction regarding the nature of their work, co-workers, supervision, and communication, all of which pertain to the human elements of the job. In contrast, they exhibited diminished satisfaction with pay, fringe benefits, contingent rewards, and promotion, as well as issues related to service organization.

These results fully confirm Herzberg’s Theory, as although intrinsic motivators increase job satisfaction, inadequate external hygiene factors limit overall satisfaction. Therefore, it is imperative to enhance employee satisfaction by strengthening both categories of factors.

Greece experienced an extensive economic crisis for a decade that weakened and made community care systems less effective, making the need for short-term and long-term measures more urgent. Short-term actions should include the introduction of fair and competitive pay structures, improvements in fringe benefits, and the implementation of supportive leadership practices. Meanwhile, long-term strategies should focus on transparent promotion systems, structured professional development and continuing education opportunities, and the provision of stable contracts to retain skilled personnel. Additionally, addressing disparities related to gender, age, and contract type could further enhance equity and staff retention. Improvements in benefits, career progression opportunities, and recognition of experience can directly influence employee morale and performance. Future research should adopt longitudinal designs to explore causal relationships, expand the sample to other regions and service types to strengthen external validity, and integrate qualitative methods to complement quantitative findings, thereby deepening understanding of employee expectations. Additionally, future studies could expand the scope of investigation by including additional psychosocial and organizational variables, such as work–life conflict or organizational commitment, to gain a broader perspective on employee well-being and performance in elder care settings.

## Figures and Tables

**Figure 1 healthcare-13-02299-f001:**
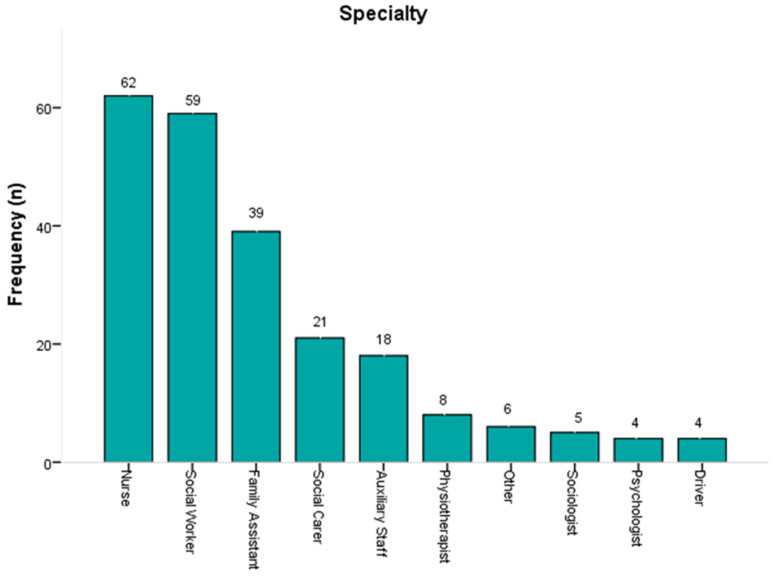
Participants’ specialty.

**Figure 2 healthcare-13-02299-f002:**
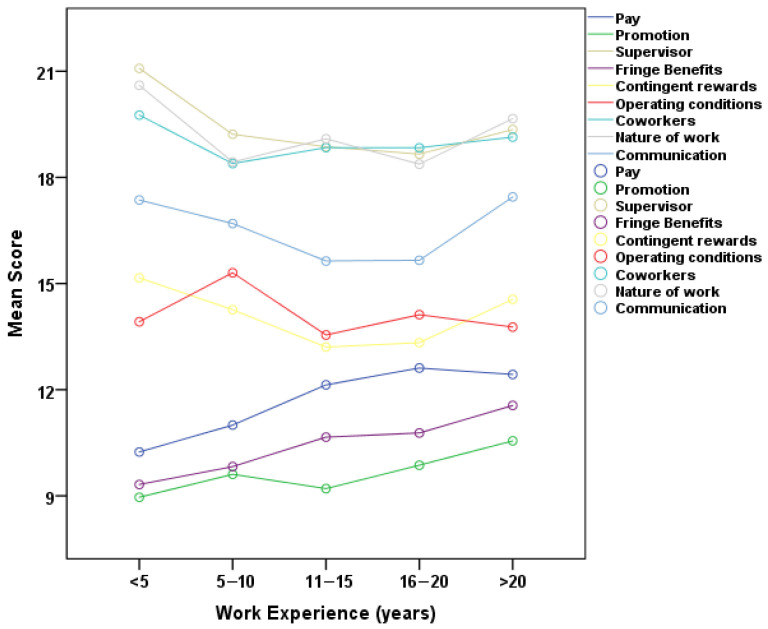
Association of JSS scales with work experience.

**Figure 3 healthcare-13-02299-f003:**
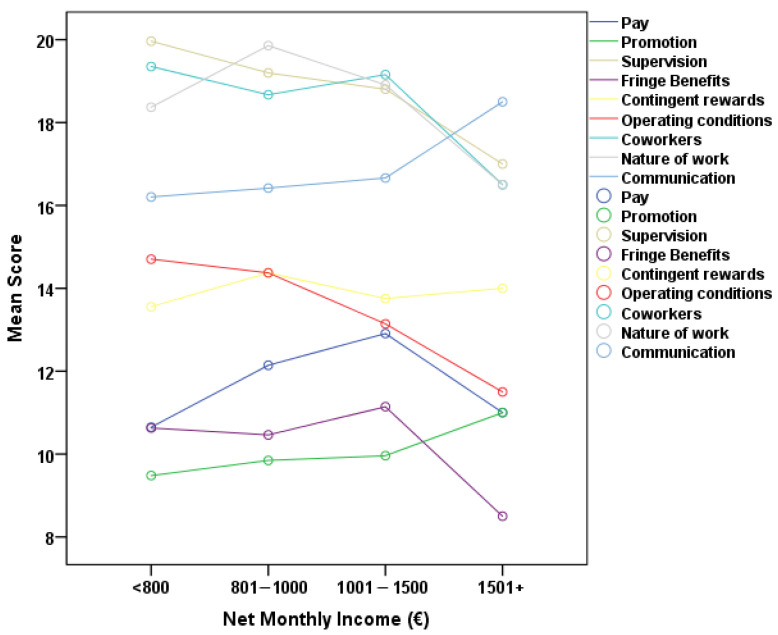
Association of JSS scales with net monthly income.

**Table 1 healthcare-13-02299-t001:** Participants’ socio-demographic profile.

Socio—Demographic Variables	Values	Ν	%
Sex	Male	25	11.%
	Female	202	89%
Education	Compulsory Education	20	8.8%
	Secondary Education	88	38.9%
	Higher Education	97	42.9%
	MSc/PhD	21	9.3%
Marital Status	Single	24	10.6%
	Married	168	74.3%
	Widowed/Divorced	34	15.0
Place of living	Rural Area	101	44.5%
	Urban Area	126	55.5%
Salary	Up to 200 €	55	24.3%
	801–1000 €	91	40.3%
	1001+ €	80	35.4%
Age groups	18–14	18	7.9%
	35–44	65	2.5%
	45–55	106	46.5%
	>56	39	17.1%
Age		Mean ± St. Deviation	Min. Max
		47.2 ± 8.4	29–64

€: Euro, the official currency of Greece

**Table 2 healthcare-13-02299-t002:** Participants’ occupational profile.

Occupational Variables	Values	Ν	%
Employment Relationship	Permanent Employment	39	17.3%
	Employment for indefinite period of time	27	11.9%
	Temporary Employment	151	66.8%
	Other	9	3.9%
Job position	Employee	179	79.2%
	Officer	47	20.8%
Years of working experience	<5 years	25	11%
	5–10 years	23	10.2%
	11–15 years	44	19.5%
	16–20 years	68	30.1%
	>20 years	66	29.2%
Community service/program	KIFI	63	27.6%
	KAPI	56	24.6%
	BSS	109	47.8%

KAPI: The Greek acronym for the open care services for the older people. KIFI: The Greek acronym for the day centers for the older people. BSS: The Greek acronym for the help at home program.

**Table 3 healthcare-13-02299-t003:** Descriptive statistics of the JSS subscales. Kolmogorov–Smirnov test (*p* < 0.05, statistically significant).

JSS Subscales	Mean ± SD	Median	Min–Max	*p* (K-S)
Pay	12.0 ± 4.6	12	4–23	0.001
Promotion	9.7 ± 4.2	10	4–22	<0.001
Supervision	19.3 ± 4.5	20	4–24	<0.001
Fringe Benefits	10.7 ± 4.4	10	4–23	0.004
Contingent Rewards	14.0 ± 4.3	14	4–24	0.003
Operating Procedures	14.0 ± 4.3	14	4–24	0.004
Coworkers	19.0 ± 3.8	19	5–24	<0.001
Nature of work	19.2 ± 4.0	20	4–24	<0.001
Communication	16.5 ± 4.7	16	5–24	0.005
Total	134.0 ± 24.0	132	36–207	0.042

*p* (K-S): Kolmogorov–Smirnov test, SD: Standard Deviation, Min–Max: Minimum–Maximum.

**Table 4 healthcare-13-02299-t004:** Effect of sex on job satisfaction. Mann–Whitney test (*p* < 0.05, statistically significant).

JSS Subscales	Male (n = 25) Mean (SD)	Female (n = 202) Mean (SD)	*p*
Pay	10.2 (4.9)	12.3 (4.5)	**0.049**
Promotion	9.2 (3.5)	9.8 (4.3)	0.591
Supervision	18.9 (3.7)	19.3 (4.6)	0.370
Fringe Benefits	8.8 (4.1)	10.9 (4.4)	**0.030**
Contingent Rewards	13.2 (3.2)	14.1 (4.4)	0.327
Operating Procedures	15.4 (4.1)	13.8 (4.2)	0.100
Coworkers	18.8 (3.2)	19.0 (3.9)	0.492
Nature of Work	18.1 (3.9)	19.3 (4.0)	0.089
Communication	16.7 (4.5)	16.5 (4.7)	0.984
Total	129.4 (19.3)	134.4 (24.5)	0.344

*p*: Mann–Whitney *p*. SD: standard deviation. Bold values indicate statistically significant results at *p* < 0.05.

**Table 5 healthcare-13-02299-t005:** Effect of age on job satisfaction: Kruskal–Wallis test (*p* < 0.05, statistically significant).

	≤45 (n = 93)		46–55 (n = 90)		56+ (n = 38)		
JSS Subscales	Mean	SD	Mean	SD	Mean	SD.	*p*
Pay	11.7	4.6	13.0	4.6	10.5	4.1	**0.008**
Promotion	9.1	4.2	10.6	4.3	9.3	3.5	**0.046**
Supervision	19.5	4.5	18.9	4.6	19.3	4.4	0.532
Fringe Benefits	10.8	4.4	11.0	4.4	9.8	4.0	0.307
Contingent Rewards	14.2	4.9	13.9	3.9	13.6	3.2	0.652
Operating Procedures	14.2	4.6	14.1	4.1	13.2	3.5	0.509
Coworkers	19.0	4.2	18.8	3.7	19.6	3.0	0.679
Nature of work	19.0	4.2	19.2	3.8	19.4	3.9	0.960
Communication	16.6	4.7	16.5	4.7	16.3	4.8	0.879
Total	134.1	26.0	135.9	21.7	130.8	17.7	0.380

JSS: Job satisfaction survey. Mean = average score; SD = standard deviation. *p*-values were calculated using the Kruskal–Wallis test to assess statistically significant differences among age groups. Bold values indicate statistically significant results at *p* < 0.05.

**Table 6 healthcare-13-02299-t006:** Effect of type of work unit on job satisfaction. Kruskal–Wallis test (*p* < 0.05, statistically significant).

	KIFI (n = 63)		KAPI (n = 56)		BSS (n = 109)		
JSS Subscales	Mean	St. Dev.	Mean	St. Dev.	Mean	St. Dev.	*p*
Pay	12.4	4.7	11.6	4.6	12.1	4.6	0.772
Promotion	9.3	3.9	9.7	3.7	10.0	4.6	0.582
Supervision	19.8	4.7	19.6	3.7	18.9	4.8	0.292
Fringe Benefits	9.7	4.3	10.4	4.6	11.4	4.2	**0.014**
Contingent Rewards	15.0	4.8	14.1	3.8	13.4	4.2	0.094
Operating Procedures	15.5	4.4	13.5	4.5	13.4	3.9	**0.008**
Coworkers	19.1	4.4	19.1	3.5	18.9	3.6	0.712
Nature of work	19.2	4.2	19.1	3.4	19.1	4.1	0.780
Communication	17.1	5.0	16.7	4.9	16.0	4.4	0.198
Total	137.3	24.9	132.0	23.8	133.1	23.6	0.554

*p*: Kruskal–Wallis. Bold values indicate statistically significant results at *p* < 0.05.

**Table 7 healthcare-13-02299-t007:** Effects on JSS subscales from demographic and occupational variables as estimated from multiple linear regression with forward selection.

JSS	Factor	Values	B	95% LL	95% UL	*p*	ANOVA P	R^2^
Pay	Age group	46–55	1.65	0.45	2.86	**0.007**	0.002	0.058
	Sex	Woman	2.32	0.42	4.23	**0.017**		
Promotion	Age group	46–55	1.30	0.19	2.41	**0.021**	0.004	0.050
	Working status	Part time	−0.70	−1.36	−0.04	**0.038**		
Supervision	Position	Head/Director	−2.18	−3.62	−0.74	**0.003**	0.003	0.039
Fringe Benefits	Working unit	BSS	1.44	0.30	2.58	**0.014**	0.014	0.027
Contingent Rewards	-	-	ΝΕ	NE	NE	**NE**	NE	NE
Operating conditions	Working position	Head/Director	−2.68	−4.02	−1.34	**<0.001**	<0.001	0.067
Coworkers	-	-	ΝΕ	NE	NE	NE	NE	NE
Nature of work	Monthly income	801–1000 €	1.26	0.20	2.32	**0.020**	0.020	0.024
Communication	-	-	ΝΕ	NE	NE	NE	NE	NE

NE: Not estimated, 95% LL: 95 Lower limit of B coefficient, 95% UL: Upper limit of B coefficient. Bold values indicate statistically significant results at *p* < 0.05.

**Table 8 healthcare-13-02299-t008:** Association of JSS total score with a list of demographic and occupational variables of respondents.

Variable		B	95% LL	95% UL	Stand Β	*p*
Sex	Woman	6.75	−3.84	17.34	0.09	0.210
Age group	46–55	0.22	−7.78	8.22	0.00	0.957
	56+	−4.62	−15.17	5.93	−0.08	0.389
Educational level	Lyceum	0.96	−12.48	14.40	0.02	0.888
	University	−3.01	−18.10	12.08	−0.07	0.695
	Master/PhD	−9.44	−28.05	9.16	−0.12	0.318
Marital status	Maried	0.40	−10.55	11.35	0.01	0.942
	Widowed/Divorced	−2.68	−16.01	10.66	−0.04	0.693
Monthly Income	801–1000 €	0.08	−9.81	9.97	0.00	0.987
	1001+ €	2.97	−9.22	15.15	0.06	0.632
Working unit	KAPI	−2.89	−13.86	8.07	−0.05	0.603
	BSS	−3.85	−12.16	4.46	−0.08	0.362
Work experience	10–15 years	−2.59	−13.96	8.79	−0.04	0.655
	15+ years	2.54	−8.64	13.72	0.05	0.654
Specialty	Other than healthcare	−4.36	−13.83	5.12	−0.09	0.366
Working status	Part time	−1.05	−6.02	3.91	−0.04	0.676
Position	Head/Director	−7.22	−15.85	1.40	−0.13	0.100

## Data Availability

The data presented in this study are available on request from the corresponding author.

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
