# Peer review of "Job Satisfaction Among Healthcare Professionals in Community-Based Care for Older People: Evidence from Greece"

_healthcare, 2025, doi:10.3390/healthcare13182299_

Round 1

Reviewer 1 Report

Comments and Suggestions for Authors
  • In the abstract, the description of the results was incomplete and lacked clear variables and comparative references. The authors were advised to revise this section for clarity.
    • The sentence, “Age appears to affect the age group,” was self-contradictory and logically tautological, offering no meaningful research insight.
    • The statement, “Job satisfaction levels were average for the three types of services,” used the term “average” ambiguously. It was unclear whether this referred to the statistical mean or to a moderate level of satisfaction. Clarification was recommended.
  • The Introduction needed to present a clearer motivation, integrate theoretical foundations, identify the research gap, and explicitly state the study objectives to highlight its policy and managerial implications.
    • The motivation for studying “home-based care for the elderly” and “job satisfaction” was fragmented. The authors were encouraged to clearly explain why these topics were important and to emphasize their policy and management relevance.
    • Although the authors mentioned factors associated with job satisfaction, they failed to focus on the central research theme. The integration of relevant theories (e.g., Herzberg’s Two-Factor Theory, JD-R Model) was recommended to explain the theoretical basis of job satisfaction and to link these frameworks to the study hypotheses.
    • The authors needed to state that most previous research had focused on hospital clinical staff, with limited attention to community care workers, and that few studies compared different types of services. This would establish a clear research gap. The manuscript should also explain how this study filled that gap and stress its implications for policy, healthcare management, and human resource allocation.
    • The study objectives should have been clearly listed (e.g., comparing job satisfaction across three service types, examining the effects of personal and work characteristics) to improve readability and structure.
  • When citing authors, only the last name should be used. The manuscript inappropriately cited the first name “Paul Spector” twice (lines 88 and 184); revisions were required.
  • The study relied solely on descriptive statistics and basic comparative analyses, which lacked analytical depth. The authors were advised to include multivariate analyses (e.g., multiple or hierarchical regression) to control for confounding variables. If the hypotheses involved interactions, moderation or mediation analyses (e.g., PROCESS macro or SEM) should be conducted. Effect sizes, collinearity diagnostics, and model explanatory power (R², ΔR²) should be reported to meet international standards of rigor.
  • The titles of Table 1 and Table 2 were identical and should be corrected.
  • Some subgroups in the results contained very small sample sizes, increasing the risk of bias and reducing statistical power. The authors were encouraged to merge similar categories, use nonparametric tests, or perform sensitivity analyses to ensure robustness. They were also advised to explicitly acknowledge this issue in the results and limitations sections and to avoid overinterpreting small subgroup findings.
  • In academic journal conventions, “Figure” should be used instead of “Diagram” for numbering conceptual models or research frameworks. The authors were advised to revise accordingly.
  • The Discussion section merely restated statistical results and lacked theoretical interpretation. The authors were encouraged to link the main findings to relevant theories (e.g., Herzberg’s Two-Factor Theory, JD-R Model) to explain how the results supported or challenged existing frameworks, thereby enhancing academic depth.
  • Policy recommendations were overly general, limited to statements such as “job satisfaction should be improved.” The authors were encouraged to propose more specific and actionable measures, such as improving compensation systems, implementing supportive leadership strategies, and providing continuing education and training. Differentiating between short-term and long-term policy recommendations would further increase the study’s practical value.
  • The study lacked specific suggestions for future research. The authors were advised to recommend longitudinal designs to validate causal relationships, to expand samples to different regions or service organizations to improve external validity, and to combine qualitative research to supplement quantitative findings, thereby deepening understanding of the topic.
  • This study only examined job satisfaction in relation to personal and job characteristics, limiting the analysis to basic variable comparisons. It failed to test interactions or mediation mechanisms between variables, and the statistical methods did not meet the rigor required by international journals. Consequently, the manuscript lacked theoretical depth and empirical innovation, resulting in limited academic contribution.
Comments on the Quality of English Language

The manuscript’s language appears overly literal in several sections, revealing traces of non-native translation. Certain expressions, such as “levels were average,” are repetitive and lack academic precision, while sentence structures are often rigid, reducing overall readability. It is recommended that the authors revise these sentences to adopt more natural and varied academic phrasing and consider professional language polishing to enhance fluency and meet the linguistic standards expected by SSCI journals.

Reviewer 2 Report

Comments and Suggestions for Authors

Dear Authors,

I appreciate the opportunity to review your manuscript and contribute to the peer-review process. I will share some suggestions that I believe may help improve your article so that it fully meets the requirements for publication in this journal.

Title

The title is unambiguous, but excessively wordy. I suggest that the authors rephrase the title to make it more concise.

  1. Abstract

Sentences that are excessively long and dense.

The use of abbreviations that have not been mentioned previously makes comprehension difficult.

This sentence already refers to the results: A total of 228 questionnaires were collected, with a response rate of 91% (lines 19-20) It should be adjusted.

The conclusion could be reworded to include more practical and political implications.

Keywords

The chosen keywords are pertinent to the study, but they do not match the indexed or Mesh terms. This choice could make it challenging to display your article when searched in the database utilizing indexed terms, decreasing its recognition and dissemination.

  1. Introduction

Line 40 - review the formatting remove the full stop

Improving the writing in general. Sometimes it appears to be merely a list of facts supported by authors, with no connection between paragraphs.

Excessive use of consecutive quotations without critical analysis.

The transition between social needs, lack of care and the relevance of professional satisfaction could be improved.

  1. Metodology

It is an extensive description, but with a scattering of ideas. Consider breaking down the different care options into separate paragraphs.

They mention the inclusion criteria, but then fail to justify the sampling. Why 250? Is it just because they met the inclusion criteria? What is the total number of people working in the centres? What is the sample size?

The text states that the scale has been translated and revised into Greek. Has it been validated? If so, please include validity indicators, such as Cronbach's alpha.

What measures were taken to protect the data collected? Did the participants sign an informed consent form, or was it only requested from the municipality?

  1. Results

The text repeats the information provided by the tables. What does this data show?

There is ambiguous use of values, with expressions such as 'significantly higher' being used without the values being presented for comparison.

  1. Discussion

It needs more theoretical support. For example, consider the stability paragraph on line 351. The assumptions made are based on the study itself. Are there no other studies?

In the next paragraph, they start again with satisfaction. However, you then start on another topic: 'nature of work'. Please consider making a more appropriate transition.

Regarding the subscale of the nature of work, it is in agreement (lines 365–366).  With what?

There is some repetition of ideas throughout the paragraphs.

There is also a lack of critical discussion about methodological limitations.

What recommendations would you make to policymakers?

Are there any limitations of the JSS? What is it?

  1. Limitations and Conclusion

Limitations are only dealt with superficially. Some kind of methodological limitation could also be addressed, such as bias in responses or generalisation.

The conclusion merely reiterates the abstract and briefly summarises the adopted policy. This section could be rewritten based on real data and the implications for both the employees and the policy.

  1. References

Standardise the way the journal is referenced. Reread the journal's guidelines.

Standardise the DOI. Some have hyperlinks, while others do not.

References 17, 20, 22, 23, 28, 29, 39, 41 and 42 have no DOI.

Ref. 24 is incomplete.

For reference 40, the year of publication is not in bold.

Based on the above assessment, I recommend a major revision, as the manuscript presents interesting and relevant data but requires substantial improvements.

Reviewer 3 Report

Comments and Suggestions for Authors

Thank you for the opportunity to review this research. I have the following comments regarding the text:
1) I believe the theoretical framework falls short in relation to elder care and caregiver satisfaction. You could add more references regarding what happens in European countries similar to the case study.
2) The case study should be noted as such within the methodology. Since Crete is an island, its unique situation may not have the same dynamics of elder care as mainland Greece.
3) I have a question regarding the KAPI and KIFI centers. Could you provide examples of the number of hours an older adult can spend in a day? Or whether these centers are attended by invitation or of their own free will. I'm not clear.
4) In no case are these centers open 24 hours or long-stay for older adults? Is this correct?
5) Regarding the statistical aspect, the management is adequate. The questionnaire used has been well validated in other contexts. Only if Cronbach's coefficient could be added to ensure reliability.
6) Since the authors only used the job satisfaction variable and mention that the majority of participants are women, it would be necessary to highlight for future research that other variables, such as work-life conflict or organizational commitment, could be measured.

Reviewer 4 Report

Comments and Suggestions for Authors

Lines 36-37: ,, Keywords: job satisfaction, elderly, open care centers, health and social care professionals, The Job Satisfaction Survey, Greece)”.

I think there are a lot of keywords.

Lines 125-126: ,, The present research was a cross-sectional quantitative study conducted between June 2022 and March 2023”.

In this case, how did you obtain the opinion from the University Ethics Committee (Protocol no. 11715/16-7-2024)?

Lines 162-163: ,, Those professionals who fulfilled the inclusion 162 criteria were 250; the final sample consisted of 228 individuals (a response rate of 91%).”

What was the age range for participants?

Lines 173-174: ,,  A self-administered questionnaire was distributed to the selected employees working in the services mentioned above.”

Is it a validated questionnaire? Or a questionnaire you designed?

Lines 344-346: ,,The majority of the sample consisted of women, a result justified by the fact that the specialties employed in all three structures of interest continue to be female dominated fields.”

Have you found this aspect in the literature?

Lines 391-394: ,, Employees aged 46-55 exhibited greater job satisfaction, especially about Pay and Promotion, in comparison to both younger and older age groups. This phenomenon may be associated with the professional stability and economic advancement experienced by individuals in this age group, who have not yet encountered retirement-related anxieties.”

Is this aspect also found in the literature?

You also mentioned the limits of the study.

Reference 24 is a small mistake

I appreciate the work done for this study.

My comments are only intended to make the paper better. Good luck!

Reviewer 5 Report

Comments and Suggestions for Authors

Reviewer’s Information

Comments and Suggestions for Authors:

The manuscript contains very interesting information but needs improvement. Please add my comments below:

Introduction

Comment 1)        

Please remove the initial point (line 40)

Comment 2)

"The introduction is quite long and could benefit from being more concise. Please summarize the key background and rationale to improve readability and focus"

Comment 3)

Please replace the term "gender" with "sex" in all manuscript, because when referring to the biological classification of individuals, the term "sex" must be used for greater precision and clarity

Comment 4)

Tables 1 and 2: The table headers, such as 'Column 1' and 'Column 2', are too generic. Could you please suggest more informative names, for example, 'Patient Demographics' and 'Treatment Outcomes', to improve clarity for the reader?

Comment 5) Table 2. Add the corresponding table footers, for example for KIFI, KAPI, BSS

Comment 6) Table 3. Define p (K-S)

Comment 7) Table 4. The headings in Table 4 are not clear, please reorganize the table and add the p-value for statistical significance in the table footer.

Comment 8) Table 5. Put the meaning of the table headers in the table footers and define St. dev. and the p-value

Comment 9) Place the p-value with statistical significance in the table captions

Comment 10) Please change the term 'diagram' to 'figure' throughout the manuscript, as 'figure' is the standard term for visual elements in scientific writing

Comment 11) Figure 1 needs to be improved. Please do the following:

a) Remove the background and margin of the figure (please leave the background white)

b) Add brighter colors to the bars

c) Add symbols to the figure caption (for example, the data shown is n or percentage). Please do not enclose numbers in boxes

d) Change the direction of the letters from bottom to top

Comment 12) Remove the gray background from the figures

Comment 13) Improve the quality of Figure 3 because it looks blurry

Additional comments :   Comment 1) Introduction: The introduction is very long and needs a better summary and structure. There are some paragraphs, such as the following: "Job satisfaction has been a problem for...", that recount various historical events, distracting the reader's attention. I suggest that the structure define the main themes and essential concepts of the manuscript and address the importance of completing the work, concluding with the overall objective.   Comment 2) Materials and Methods: Divide the participant section and clearly state the inclusion and exclusion criteria, how they were recruited, and the sample size calculation.   Comment 3) Materials and Methods: Describe in a separate section the information corresponding to the 3 included services: KAPI (Open Care Centers for the Elderly), KIFI (Day Care Centers for the Elderly), and the Help at Home Program (BSS).   Comment 4) Table 5) What do the numbers 93, 90, and 38 in parentheses refer to? Indicate the column headings and table footers. The information is confusing.   Comment 5) Tables) Give specific names to each column in the tables and corresponding table captions, specifying abbreviations, statistical significance, and how the data is expressed by variable type, for example n and percentage.
